# Rewrite Caption Semantics: Bridging Semantic Gaps for Language-Supervised Semantic Segmentation

Yun Xing[1]    Jian Kang[1]    Aoran Xiao[1]    Jiahao Nie[1]    Ling Shao[2]    Shijian Lu[1]*

[1] Nanyang Technological University
[2] UCAS-Terminus AI Lab, UCAS, China

## Abstract

Vision-Language Pre-training has demonstrated its remarkable zero-shot recognition ability and potential to learn generalizable visual representations from language supervision. Taking a step ahead, language-supervised semantic segmentation enables spatial localization of textual inputs by learning pixel grouping solely from image-text pairs. Nevertheless, the state-of-the-art suffers from clear *semantic gaps* between visual and textual modality: plenty of visual concepts appeared in images are missing in their paired captions. Such semantic misalignment circulates in pre-training, leading to inferior zero-shot performance in dense predictions due to insufficient visual concepts captured in textual representations. To close such *semantic gap*, we propose Concept Curation (CoCu), a pipeline that leverages CLIP to compensate for the missing semantics. For each image-text pair, we establish a *concept archive* that maintains potential visually-matched concepts with our proposed *vision-driven expansion* and *text-to-vision-guided ranking*. Relevant concepts can thus be identified via *cluster-guided sampling* and fed into pre-training, thereby bridging the gap between visual and textual semantics. Extensive experiments over a broad suite of 8 segmentation benchmarks show that CoCu achieves superb zero-shot transfer performance and greatly boosts language-supervised segmentation baseline by a large margin, suggesting the value of bridging *semantic gap* in pre-training data. Code is available at `https://github.com/xing0047/rewrite`.

## 1 Introduction

Vision-Language Pre-training [34, 21, 1, 10], which aims to learn visual representations directly from natural language supervision, has endowed existing recognition systems with superior generality and open-vocabulary understanding capability. As a representative, CLIP [34] performs contrastive language-image pre-training on 400M web-crawled image-text pairs, whereby the learnt models may effortlessly transfer to a wide spectrum of classification tasks in a zero-shot manner. Motivated by the breakthrough, recent studies [43, 35, 44] extend the supervision paradigm to semantic segmentation, enabling spatial localization of textual queries in images and pixel grouping with solely supervision from image-text pairs. Distinct from conventional semantic segmentation, the language-supervised paradigm obviates the need for costly manual pixel-level annotation and enables million-level pre-training scale with much less effort.

Despite the progresses [43, 35] in language-supervised semantic segmentation, the pre-training stage still suffers heavily from clear *semantic gap* between visual and textual modality. In image-text pairs used for pre-training, it is ubiquitous that visual concepts appeared in images are missing in the corresponding textual captions. This happens largely because captions merely describe salient concepts that are worthy of mention [16, 25], while naturally forgo full semantic coverage of images

---

*Corresponding Author.

37th Conference on Neural Information Processing Systems (NeurIPS 2023).

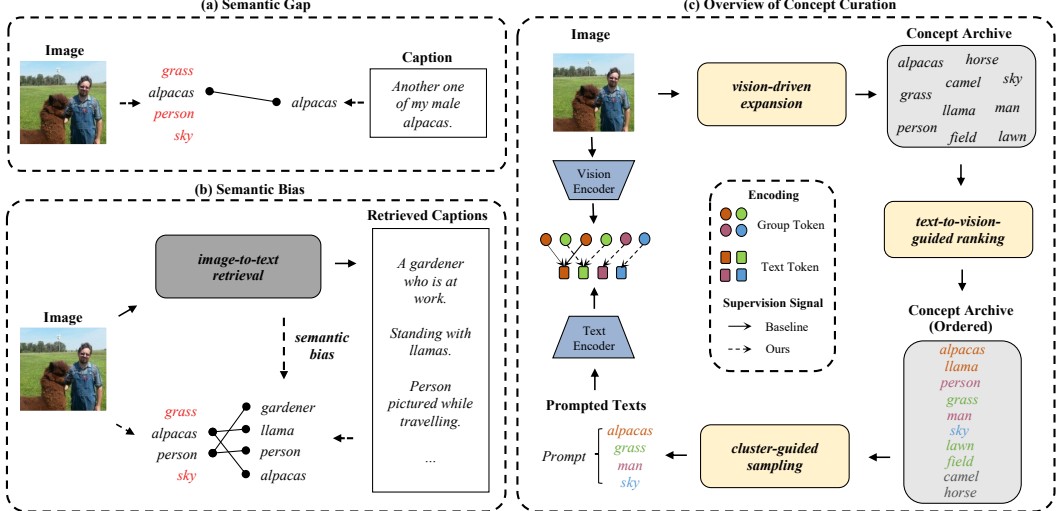

Figure 1: Cross-modal *semantic gap* is prevalent in web-crawled image-text pairs. As in (a), the caption text often captures certain salient visual concepts only in the paired image but misses many others (i.e., '*person*', '*grass*', and '*sky*') that are also useful in image-text modeling. Leveraging CLIP [34], more useful visual concepts could be captured via image-to-text retrieval, but the retrieved captions usually suffer from the *semantic bias* as in (b) (i.e., '*person*' recovered but '*grass*' and '*sky*' still missing). Our proposed Concept Curation (CoCu) bridges the cross-modal *semantic gap* effectively by *vision-driven expansion*, *text-to-vision-guided ranking* and *cluster-guided sampling* while avoiding the negative effect by *semantic bias*, as illustrated in (c). Best viewed in color.

(Fig. 1 (a)). Under the presence of clear cross-modal *semantic gap* in image-text pairs, the pre-training stage of language-supervised segmentation is found to be harder to converge, leading to inferior zero-shot performance on downstream tasks (more details are elaborated in Section 4.3).

This work explores to bridge *semantic gaps* in language-supervised semantic segmentation. For each image in the pre-training data, the goal is to recover the missing visual concepts in its paired caption for more comprehensive image-text modeling. With the rich vision-language correlations in off-the-shelf foundation models such as CLIP [34], a straight solution is to retrieve the missing concepts from the text captions of pre-training data. However, such retrieved captions suffer from the *semantic bias* illustrated in Fig. 1 (b) (e.g., "person" recovered but "grass" and "sky" still missing). The root cause lies with the original text captions in foundation model pre-training, which only capture salient concepts appeared in the paired images. Hence, the retrieved captions and concepts still suffer from clear cross-modal *semantic gap*.

We propose Concept Curation (CoCu), a novel pipeline that side-steps the negative effect by *semantic bias*[2] while exploiting vision-language foundation models for semantic segmentation. CoCu consists of three sequential stages: 1) *vision-driven expansion* that constructs *concept archive* via cross-image retrieval; 2) *text-to-vision-guided ranking* that scores the retrieved concepts according to their assigned relevancies; and 3) *cluster-guided sampling* that exploits semantic diversity beyond the relevancy scores for concept ranking (Fig. 1 (c)). We perform pre-training from scratch on the segmentation backbone of [43] and evaluate zero-shot transfer over 8 widely adopted segmentation benchmarks. The experiments show that the proposed CoCu improves the baseline as well as the state-of-the-art consistently by large margins, indicating the necessity of closing the *semantic gap* and the effectiveness of our designs in concept archiving and concept ranking.

In summary, the contributions of this work are three-fold. *First*, we identify the issue of *semantic gap* in language-supervised semantic segmentation, and demonstrate the effectiveness of mining more relevant visual concepts for segmentation model pre-training. *Second*, we design Concept Curation (CoCu), a novel pipeline that constructs *concept archives* to expand and identify relevant

---

[2]to clarify, we refer to *semantic gap* as a problem in web-crawled image-text pairs and *semantic bias* as an issue in pre-trained vision-language model.

visual concepts from pre-training data, mitigating the *semantic gap* between textual and visual modalities effecly. *Third*, extensive experiments show that the proposed pipeline achieves superior zero-shot transfer performance and outperforms the state-of-the-art across 8 segmentation benchmarks consistently by large margins.

## 2 Related Works

**Semantic Segmentation.** Partitioning an image into semantic regions, also known as semantic segmentation, has been widely studied due to myriad real-world applications such as video surveillance and autonomous driving. It has been explored along different directions, e.g., by designing different network architectures [29, 9, 11, 42], constructing benchmarks of diverse scenes and categories [12, 50, 5, 17], etc. However, collecting per-category dense annotations for supervised training is notoriously labor-intensive, which impedes the upscaling of semantic vocabulary greatly. Different from conventional semantic segmentation, segmentation from language supervision [43, 28, 7, 32, 35, 44] relieves the burden of mask annotation by leveraging image-text pairs available on the Internet. Beyond that, it can handle arbitrary new semantics thanks to the language supervision paradigm, making it feasible to learn generalizable segmentation models.

**Vision-Language Pre-training.** Recently, Vision-Language Pre-training has become a predominant trend by learning visual representations from natural language supervision [34, 21, 24, 47, 1, 31, 10, 49]. By matching billion-scale image-text pairs via contrast, the learnt representations can be seamlessly transferred to various downstream classification tasks in a zero-shot manner. As a representative, CLIP [34] can match the performance of supervised baselines on ImageNet [13], meanwhile obtain competitive performance over plethora of downstream tasks without accessing any target data. The same learning paradigm has recently been explored for the task of semantic segmentation by hierarchical grouping [43], supervision mining [35, 44, 7, 32], etc. Nevertheless, state-of-the-art language-supervised segmentation is held back by the cross-modal *semantic gap* between textual and visual pre-training data. Instead of relaxing the strict one-to-one correspondence in vanilla contrastive learning [35], we mitigate the *semantic gap* by automated curation of relevant visual concepts through concept expanding and concept ranking.

**Open-Vocabulary Semantic Segmentation.** Open-Vocabulary Semantic Segmentation has been studied extensively and most existing work can be broadly grouped into three categories. **Mix Supervision:** the first category follows a zero-shot manner [4, 23] which aims to segment new classes by learning from densely annotated seen classes in the pre-training data. Recently, several studies [18, 22, 46, 19, 26, 52, 45, 48] introduce language supervision to enhance the generality of the learnt zero-shot models. These approaches require no data annotation from new classes, but still rely on dense annotations from seen classes during the pre-training stage. **No Supervision**: the second category follows a training-free approach [51, 38, 37] which explores the segmentation potential of frozen vision-language models (VLMs) [51, 38, 37] to predict segmentation masks. However, most VLMs are trained with image-level supervision which restricts their capability on pixel/region-level predictions in semantic segmentation. **Language Supervision:** the third category follows a pure-language-supervision paradigm [43, 28, 32, 35, 7, 44] which aims to learn pixel grouping from solely image-text pairs. Our work follows the third approach. Different from existing studies, we identify the *semantic gap* in pre-training image and text data and design concept curation that mitigates the *semantic gap* with clearly improved semantic segmentation performance, more details to be described in the ensuing subsections.

## 3 Methodology

With clear *semantic gaps* between visual and textual concepts in pre-training data as illustrated in Fig. 1 (a), one naïve solution is to employ given image as query to retrieve related captions and derive missing textual concepts as described in Sec. 3.2. However, the naïve solution suffers from clear *semantic bias* as most VLM-retrieved captions contain salient concepts only as illustrated in Fig. 1 (b). We thus further design *vision-guided expansion*, *text-to-image-guided ranking* and *cluster-guided sampling* for better mitigation of the *semantic gap* as presented in Fig. 1 (c) and Sec. 3.3.

## 3.1 Revisiting GroupViT

**Segmentation Architecture.** We use GroupViT [43] as the segmentation backbone for pre-training. Assume a batch of image-text pairs $\{(x^I, x^T)\}_{i=1}^B$, where $x^I$ amd $x^T$ denotes an image and its paired caption, respectively. For the vision flow, a grouping-aware transformer $\mathcal{F}_s^I$ encodes image $x^I$ as G segment tokens $z_{seg}^I = \{z_{seg_g}^I, g = 1, ..., G\} \in \mathbb{R}^{G \times d}$, where each segment token $z_{seg_g}^I \in \mathbb{R}^d$ encodes an arbitrary-shaped region in image $x^I$.

**Image-Caption Contrastive Loss.** To perform pre-training, the segment tokens $Z_{seg}^I$ are merged via average pooling, producing a global representation $z_{seg}^I \in \mathbb{R}^d$ that captures all the visual concepts appeared in image $x^I$. Meanwhile, the paired caption $x^T$ is encoded to $z^T \in \mathbb{R}^d$ by a text encoder $\mathcal{F}^T$. The visual embedding $z_{seg}^I$ and textual embedding $z^T$ are mapped to the same space by separate linear projectors. The segmentator is then learnt from language supervision by the standard contrastive objective InfoNCE [33], which is defined as:

$$\mathcal{L}_{I \to T} = -\frac{1}{B} \sum_{i=1}^B \log \frac{\exp(z_i^I \cdot z_i^T / \tau)}{\sum_{j=1}^B \exp(z_i^I \cdot z_j^T / \tau)} \tag{1}$$

$$\mathcal{L}_{T \to I} = -\frac{1}{B} \sum_{i=1}^B \log \frac{\exp(z_i^T \cdot z_i^I / \tau)}{\sum_{j=1}^B \exp(z_i^T \cdot z_j^I / \tau)} \tag{2}$$

where $\tau$ is a learnable parameter initialized with 0.07 [34] and the $z^I \cdot z^T$ computes the cross-modal cosine similarity.

**Multi-Label Loss**. Beyond learning segmentation from raw caption $x^T$, GroupViT [43] further introduces $L$ extra text labels $\{x^{T_l}, l = 1, ..., L\}$ by prompting the extracted concepts $\{c_l, l = 1, ..., L\}$ with handcrafted templates [34] (e.g., "a photo of a {concept}"). The $L$ text labels are fed to the same text encoder $\mathcal{F}^T$ to obtain textual representations of $\{z^{T_l}, l = 1, ..., L\}$. The language supervision by multi-label loss is thus defined as:

$$\mathcal{L}_{I \to \{T_l\}_{l=1}^L} = -\frac{1}{B} \sum_{i=1}^B \log \frac{\sum_{l=1}^L \exp(z_i^I \cdot z_i^{T_l} / \tau)}{\sum_{l=1}^L \sum_{j=1}^B \exp(z_i^I \cdot z_j^{T_l} / \tau)} \tag{3}$$

$$\mathcal{L}_{\{T_l\}_{l=1}^L \to I} = -\frac{1}{LB} \sum_{l=1}^L \sum_{i=1}^B \log \frac{\exp(z_i^{T_l} \cdot z_i^I / \tau)}{\sum_{j=1}^B \exp(z_i^{T_l} \cdot z_j^I / \tau)} \tag{4}$$

The overall training objective of learning segmentation from language supervision in [43] is defined as:

$$\mathcal{L} = \mathcal{L}_{I \leftrightarrow T} + \mathcal{L}_{\{T_l\}_{l=1}^L \leftrightarrow I} \tag{5}$$

**Discussion.** We use the exact same training objective as in GroupViT [43] to learn segmentation from language supervision. For each pre-training image $x^I$, the *multi-label loss* enhances the contrastive learning [43] with $L$ extra positive pairs and $L(B-1)$ extra negative pairs ($B$ denotes batch size used in pre-training). However, we highlight that the simple concept prompting does not expand the textual concepts much. The cross-modal *semantic gap* still exists and circulates in pre-training, which holds back the training convergence and degrades the zero-shot transfer.

## 3.2 Naïve Solution

**Caption Curation.** The web-crawled image-text pairs are often noisy with imprecise and even irrelevant text descriptions [41]. In addition, many visual concepts (especially those inconspicuous in the background) in images are often missing in the corresponding text descriptions. Both factors lead to clear *semantic gaps* between web-crawled images and texts. With super-rich image-text correlations in pre-trained VLMs such as CLIP [34], a straight solution, which we term by *caption curation*, is to apply $x^I$ as query to retrieve $L$ extra captions $\{x^{T_l}, l = 1, ..., L\}$ from pre-training data. The *semantic gaps* between visual and textual modality could thus be mitigated by identifying relevant concepts from the retrieved captions.

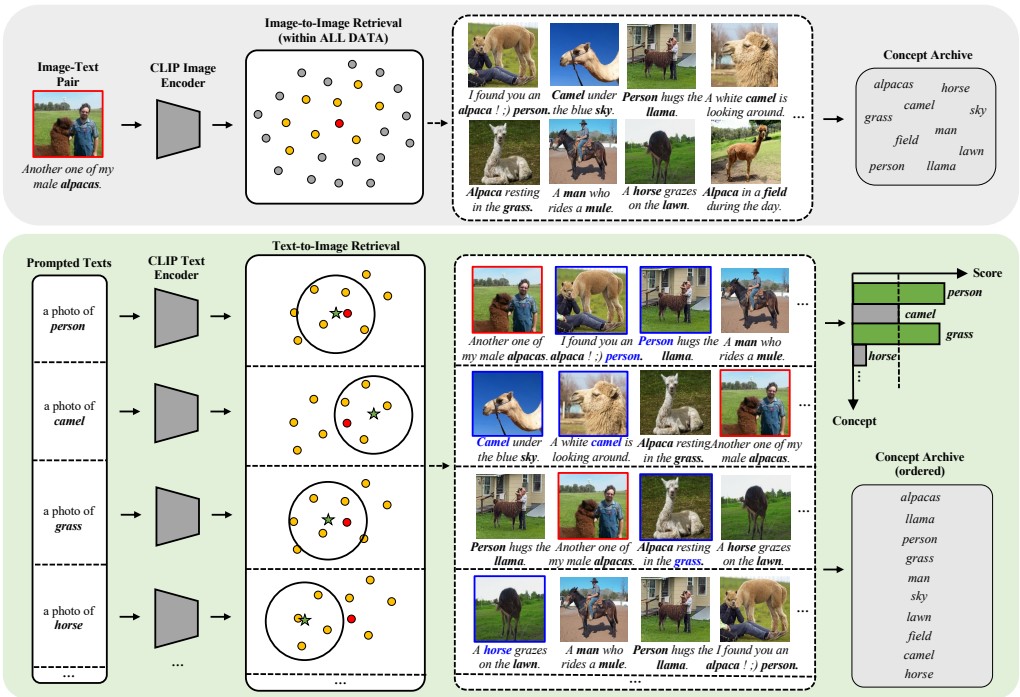

Figure 2: Illustration of *vision-driven expansion* (above) and *text-to-image-guided ranking* (below) in CoCu. To compensate for missing semantics, *vision-driven expansion* establishes an archive of potential matched concepts through image-to-image retrieval, while *text-to-vision-guided ranking* scores retrieved concepts based on assigned relevancy. The textual concepts can later be identified in pre-training by sampling. In the figure, images with a blue border □ are retrieved via expanded concepts (marked as blue) using their paired captions, while images with a red border □ represent images for curation (as anchor). Best viewed in color.

**Semantic Bias.** Though *caption curation* expands visual concepts effectively, retrieved captions often suffer from clear *semantic bias*: VLMs tend to retrieve salient concepts but miss many inconspicuous ones that are also useful in image description. Consequently, visual concepts $C^I = \{c_l, l = 1, ..., M^I\}$ appeared in an image $x^I$ are usually clearly more than textual concepts $C^T = \{c_l, l = 1, ..., M^T\}$ extracted from $\{x^T, x^{T_1}, ..., x^{T_L}\}$ (i.e., $M^I > M^T$). The root cause of the *semantic bias* lies with the loose correlation between the visual and textual pre-training data of VLMs, where most captions just capture partial visual concepts appeared in the paired images [25]. The *semantic bias* thus impedes convergence and effectiveness of language-supervised training without language supervision available for those non-described image regions.

### 3.3 Concept Curation

To bridge semantic gaps in image-text pairs, we propose **Concept Curation (CoCu)** to rewrite caption semantics with the help of a pre-trained vision-language model. Consequently, **CoCu** finds more concept candidates that are aligned to images and compensate for missing semantics in captions. In pre-training, a multi-modal segmentor matches images and visual-enriched captions by contrastive objectives mentioned in Sec. 3.1, encoding better vision-language alignment in its representations. Details of CoCu are described as below.

**Vision-driven Expansion.** For an image-text pair $(x^I, x^T)$, the goal of the vision-driven expansion is to build an archive of textual concepts $C^T = \{c_m, m = 1, ..., M\}$ that are potentially matched with $x^I$ as illustrated in Fig. 2. Instead of acquiring $C^T$ via direct text retrieval as in *caption curation*, we resort to cross-image retrieval to achieve the expansion. Concretely, $N$ image-text pairs $P = \{(x_i^I, x_i^T), i = 1, ..., N\}$ are automatically selected from the pre-training data, where $\{x_i^I, i = 1, ..., N\}$ are N captioned images whose visual features match the best with that of $x^I$ (all encoded by CLIP). $C^T$ can thus be derived by extracting textual concepts from the captions

$\{x_i^T, \ i = 1, ..., N\}$ of the $N$ best matched images. Compared with the *caption curation* that retrieves $L$ descriptions $\{x_i^T, \ i = 1, ..., L\}$, *vision-driven expansion* exploits all visual information in images instead of biased caption texts (mostly describes salient visual concepts only), which helps restore more relevant textual concepts. In addition, it builds an extra image set $\{x_i^I, \ i = 1, ..., N\}$ that plays a pivotal role in the upcoming stages.

**Text-to-Vision-Guided Ranking.** For each textual concept $c_m$ in the concept archive $C^T$, we assign a score $s_{c_m}$ to represent its relevancy to image $x^I$. A naïve solution to $p_m$ is to compute the cosine similarity between the visual representation of $x^I$ and textual representation of $t_m$ encoded by CLIP [34], which is simply defined as:

$$s_{c_m}^a = f(x^I, \ t_m) \tag{6}$$

where $t_m$ is derived from visual concept $c_m$ via prompt engineering [34, 43]. Note direct text retrieval could easily get biased towards salient visual concepts here, imposing low relevance scores for other text concepts in *concept archive*. Beyond $f(x^I, \ t_m)$, we also design a non-biased metric to capture the relevancy between image $x^I$ and concept $c_m$. Specifically, for the $N$ retrieved image-text pairs $P$, we first extract a subset $P_G$ (blue box/text in Fig. 2 (below)), whose caption of each image-text pair contains the textual concept $c_m$. The non-biased metric is thus defined with $x^I$ (red box) and image-text pairs $P_G = \{(x_i^I, x_i^T), \ i = 1, ..., N'\}$ as follows:

$$s_{c_m}^b = \frac{(1 + N')f(t_m, x^I)}{f(t_m, x^I) + \sum_{i=1}^{N'} f(t_m, x_i^I)} \tag{7}$$

The given term functions as follows: 1) lower relevancy between image $x^I$ and irrelevant concepts (e.g., 'horse' in Fig. 2); 2) enhance relevancy between $x^I$ and inconspicuous concepts (e.g., 'grass'). Instead of computing relevancies of $x^I$ to all textual concepts in a single run, we consider one $c_m$ at a time and measure its relevancy by comparing $f(x^I, \ t_m)$ with $\{f(x_i^I, \ t_m), \ i = 1, ..., N'\}$. The idea behind this is simple: 1) comparing the responses of images $\{x^I, x_1^I, ..., x_{N'}^I\}$ to the same textual concept $c_m$; 2) high relevancy is given if response of $x^I$ to textual concept $c_m$ is comparably high and vice versa. Take the visual concept 'grass' in Fig. 2 (below) as an example. The $t_m$ ('a photo of grass') causes $x^I$ to rank considerably higher than image captioned with $c_m$ ('grass'). In this case, we should be fairly confident to pass the concept $c_m$ to $x^I$. In conclusion, the relevancy score is simply defined as:

$$s_{c_m} = s_{c_m}^a + s_{c_m}^b \tag{8}$$

we perform ranking according to computed relevancies $\{s_{c_m}, \ m = 1, ..., M\}$, which represents chances identified by later sampling.

**Cluster-guided Sampling.** The pre-training can thus be empowered by including the expanded and ranked textual concepts which as selected by sampling $L$ textual concepts according to their computed relevancies as in [43]. However, selection with relevancy alone is often short of semantic diversity in the selected text concepts. Instead of directly selecting $L$ concepts from the ranked archive $C^T$, we partition $C^T$ into $L$ semantic clusters based on their textual representations and sample one textual concept from each semantic cluster. The *cluster-guided sampling* has two clear benefits: 1) it includes more diverse semantics in each single training step; 2) it keeps good consistency with the expression of visual concepts, more details to be discussed in Sec. 4.4 and appendix.

## 4 Experiments

### 4.1 Experimental Setup

**Training Detail**. We follow the prior study [43] and conduct pre-training on three publicly available image-text datasets: CC3M (C3) [36], CC12M (C12) [8], YFCC14M (Y14) [39]. For fair comparison, we use the same GroupViT [43] as the visual encoder, which is built upon ViT-S backbone [14, 40] and learnt from scratch. We set the global batch size for contrastive learning as 1,024 and use 4 Tesla V100 GPUs to carry out pre-training for all experiments. Consistent with [43], we set the initial learning rate to 0.0016. The pre-training undergoes 30 epochs, with a linear warmup for the first 2 epochs and a cosine schedule for the remaining epochs. $L$ is set to 3. In our ablations and discussions, we report the performance of models pre-trained on CC3M.

**Implementation of curation.** The curation pipeline utilizes clip-retrieval [3], a utility that enables efficient computation of CLIP embeddings and fast indexing for retrieval. We employ the CLIP ViT-B/16 [34] model for image/text inference and concept curation. For efficient semantic searching, we build indexing systems using autofaiss [3]. It is worth mentioning that alternative systems can also be used for implementing the curation process.

**Evaluation.** We benchmark zero-shot transfer performance of CoCu on the validation splits of eight different datasets that cover a myriad of scenes and category sets, including Pascal VOC [15], Pascal Context [30], COCO [27], ImageNet-S-50, ImageNet-S-300 [17], COCO Stuff [5], Cityscapes [12], and ADE20K [50]. For the first five datasets, we follow [43] and evaluate foreground classes by thresholding the similarity between visual and textual embeddings. For other datasets, we evaluate both foreground and background classes. More details are given in the appendix.

## 4.2 Comparison with the state-of-the-art

We first benchmark CoCu with state-of-the-art zero-shot methods [51, 43] and evaluate its effectiveness. Specifically, we follow prior work [43] and pre-train CoCu over the combination of C3, C12, and Y14 datasets. Tab. 1 reports zero-shot segmentation results. Besides GroupViT as the baseline method, we also compare the advanced MaskCLIP [51]), which directly leverages the frozen CLIP model for segmentation prediction without pre-training. In addition, for a comprehensive comparison, we list the performance of other advanced methods including 1) fully-supervised method [40] that provides Oracle's performance, 2) self-supervised methods [20, 6] that pre-train models with unlabeled data and fine-tuning models over segmentation datasets. Detailed implementations of the comparing methods could be found in the appendix.

As shown in Tab. 1, MaskCLIP achieves limited segmentation performance, primarily due to CLIP being trained with image-level supervision and thus falling short in precise pixel-level predictions. GroupViT achieves better performance than MaskCLIP, but still limited by insufficient supervision from language side in pre-training. On the contrary, our CoCu achieves the best segmentation performance over all eight benchmarks, surpassing GroupViT by large margins on average. This indicates the necessity of bridging *semantic gaps* in language-supervised semantic segmentation and the effectiveness of our design.

Table 1: **Performance of different zero-shot methods for semantic segmentation.** Abbreviations of benchmarks, from left to right: Pascal VOC [15], Pascal Context [30], Microsoft COCO [5], ImageNet-S [17], Cityscapes [12], and ADE20K [50]. BS denotes pre-training batch size, while LC represents local consistency [2] in mask prediction. † denotes our re-implementation. CoCu consistently achieves the best performance across all benchmarks.

| Method | Pretrain Data | Supervision | LC | BS | Backbone | PVOC | PCON | COCO | IN50 | IN300 | CITY | ADE | STUF | AVG |
|---|---|---|---|---|---|---|---|---|---|---|---|---|---|---|
| DeiT [40] | IN-1K | full | | - | ViT-S | 53.0 | 35.9 | - | - | - | - | - | - | - |
| MoCo [20] | IN-1K | self | | - | | 34.3 | 21.3 | - | - | - | - | - | - | - |
| DINO [6] | IN-1K | self | | - | | 39.1 | 20.4 | - | - | - | - | - | - | - |
| MoCo [20] | C12,Y14 | self | | - | | 36.1 | 23.0 | - | - | - | - | - | - | - |
| DINO [6] | C12,Y14 | self | | - | | 37.6 | 22.8 | - | - | - | - | - | - | - |
| MaskCLIP [51] | - | N.A. | ✓ | - | ResNet-50 | 41.5 | 18.5 | 10.5 | 13.8 | 7.9 | 18.8 | 8.3 | 10.2 | 15.0 |
| MaskCLIP [51] | - | N.A. | ✓ | - | ViT-B/16 | 49.5 | 21.7 | 13.6 | 25.9 | 11.7 | 19.8 | 9.5 | 12.5 | 20.5 |
| GroupViT [43] | C3,C12,Y14 | text | | 4,096 | ViT-S | **52.4** | 22.3 | **24.3** | 44.3 | 23.5 | 15.8 | 10.4 | 13.0 | 25.7 |
| GroupViT† [43] | C3,C12,Y14 | text | | 1,024 | ViT-S | 43.8 | 19.3 | 19.6 | 37.8 | 17.2 | 17.2 | 10.4 | 13.6 | 22.4 |
| CoCu (ours) | C3,C12,Y14 | text | | 1,024 | ViT-S | 49.7 | 22.8 | 22.0 | 46.7 | 24.7 | 21.9 | 12.0 | 14.9 | 26.8 |
| GroupViT† [43] | C3,C12,Y14 | text | ✓ | 1,024 | ViT-S | 45.4 | 19.9 | 20.3 | 39.2 | 17.7 | 17.6 | 10.6 | 13.9 | 23.1 |
| CoCu (ours) | C3,C12,Y14 | text | ✓ | 1,024 | ViT-S | 51.4 | **23.6** | 22.7 | **48.8** | **25.5** | **22.1** | **12.3** | **15.2** | **27.7** |

We further evaluate the robustness of CoCu with different pre-train data. Spcifically, we pre-train GroupViT and CoCu over CC3M and CC12M, respectively. We also sub-sample half of image-text pairs from CC12M (denoted as C12*) for pre-training. Tab. 2 shows the experimental results. We can observe consistent yet significant performance gains on eight benchmarks. The improvement by bridging *semantic gap* is thus robust and not affected by pre-training size.

---

[3] https://github.com/criteo/autofaiss.git

Table 2: **Zero-shot semantic segmentation performance with different pre-training data.** CoCu consistently outperforms the baseline method GroupViT across all benchmarks, demonstrating its effectiveness in bridging *semantic gaps* and achieving significant improvements.

| Method | Pretrain Data | PVOC (21) | PCON (60) | COCO (81) | IN50 (51) | IN300 (301) | CITY (19) | ADE (150) | STUF (171) | AVG |
|---|---|---|---|---|---|---|---|---|---|---|
| GroupViT[†] [43] | C3 | 15.5 | 10.4 | 6.5 | 10.2 | 2.9 | 8.1 | 4.4 | 7.7 | 8.2 |
| CoCu | C3 | 30.6 (+15.1%) | 13.9 (+3.5%) | 10.8 (+4.3%) | 19.3 (+9.1%) | 7.3 (+4.4%) | 8.2 (+0.1%) | 6.1 (+1.7%) | 8.5 (+0.8%) | 13.1 (+4.9%) |
| GroupViT[†] [43] | C12* | 32.9 | 13.3 | 12.9 | 27.9 | 12.4 | 10.7 | 5.6 | 8.6 | 15.5 |
| CoCu | C12* | 34.1 (+1.4%) | 16.4 (+3.1%) | 17.0 (+4.1%) | 33.0 (+5.1%) | 17.3 (+4.9%) | 11.8 (+1.1%) | 8.1 (+2.5%) | 9.5 (+0.9%) | 18.4 (+2.9%) |
| GroupViT[†] [43] | C3,C12* | 36.5 | 15.9 | 16.2 | 33.5 | 14.0 | 12.4 | 7.0 | 10.5 | 18.2 |
| CoCu | C3,C12* | 38.1 (+1.6%) | 19.2 (+3.3%) | 20.1 (+3.9%) | 35.8 (+2.3%) | 18.9 (+4.9%) | 14.7 (+2.3%) | 9.5 (+2.5%) | 11.4 (+0.9%) | 21.0 (+2.8%) |
| GroupViT[†] [43] | C12 | 37.5 | 18.0 | 18.3 | 35.7 | 16.6 | 13.5 | 9.1 | 13.1 | 20.2 |
| CoCu | C12 | 40.9 (+3.4%) | 21.2 (+3.2%) | 20.3 (+2.0%) | 40.0 (+4.3%) | 19.4 (+2.8%) | 15.0 (+1.5%) | 11.1 (+2.0%) | 13.6 (+0.5%) | 22.7 (+2.5%) |

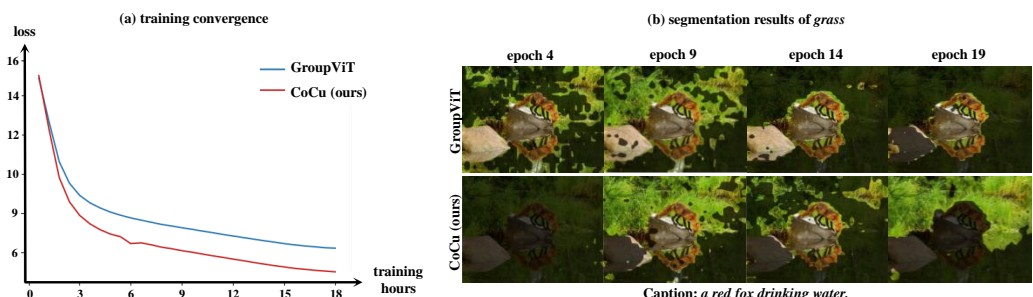

Figure 3: **CoCu enhances training convergence.** (a) The training loss curves of GroupViT and CoCu demonstrate that CoCu significantly accelerates pre-training convergence. (b) CoCu achieves superior binary segmentation results (second row) compared to GroupViT (first row) for the concept of "grass," which is missing in the caption, using an example image captioned as "a red fox drinking water." Best viewed in color.

## 4.3   CoCu helps convergence

**Loss Curve.** In Figure 3 (a), we compare the pre-training loss curves of GroupViT and our proposed method CoCu. We can see that CoCu exhibits a notably faster convergence rate, primarily attributed to the inclusion of curated semantic concepts for each image, resulting in more effective contrastive learning. Additionally, CoCu achieves a lower minimum loss by extracting significantly richer language concepts from image data. This enriches the training process by incorporating more identified image regions and ultimately learning representations that better align with the training data.

**Qualitative Comparison**. We also present qualitative results that demonstrate the effectiveness of CoCu. In Figure 3 (b), we show binary segmentation results of GroupViT (first row) and CoCu (second row) over an example image with the caption "a red fox drinking water." Our focus is on the concept of "grass," which is missing in the caption. We compare the visual responses of models trained using these two methods at different checkpoints. Both methods improve progressively during training. However, GroupViT fails to correctly localize the region of "grass" due to the lack of direct supervision from the language side. In contrast, CoCu bridges the semantic gap by accurately capturing and localizing "grass," encoding it in representations during pre-training. Consequently, it achieves significantly better segmentation results under zero-shot context.

Figure 4 displays the activation maps of GroupViT and CoCu for different concepts as text inputs that do not appear in the corresponding captions. These maps further demonstrate the superiority of CoCu in language-supervised learning. In all presented images, GroupViT incorrectly activates corresponding regions based on the given text inputs (e.g., activating the "sky" region with a text input of "person" for the first image). In contrast, CoCu enables the segmentor to have the highest activations on visually relevant regions indicated by the text. This suggests that segmentors derived from our method have a better capability to discriminate various visual concepts. More convergence results can be found in the appendix.

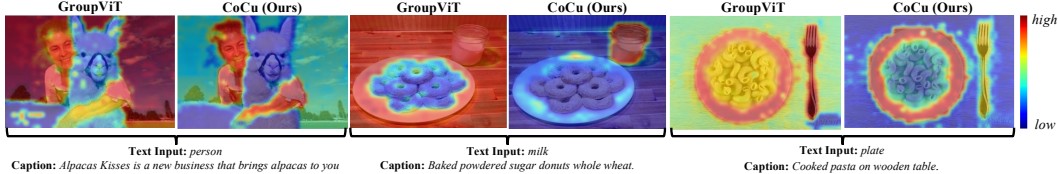

| GroupViT | CoCu (Ours) | GroupViT | CoCu (Ours) | GroupViT | CoCu (Ours) | |
|---|---|---|---|---|---|---|
| **Text Input:** *person* | | **Text Input:** *milk* | | **Text Input:** *plate* | | *high* |
| **Caption:** *Alpacas Kisses is a new business that brings alpacas to you* | | **Caption:** *Baked powdered sugar donuts whole wheat.* | | **Caption:** *Cooked pasta on wooden table.* | | *low* |

Figure 4: **Visualization of activation heatmaps.** GroupViT fails to activate on corresponding visual regions for concepts not represented in captions, while CoCu exhibits significantly better localization. High activation is shown as red, and low activation is displayed as blue. Best viewed in color.

Table 3: **Ablation study of CoCu.** We conduct an ablation study on each designed modules. Zero-shot transfer performance on semantic segmentation results are reported, averaged across eight evaluation datasets. "Naïve ranking" refers to solely using cosine similarity between visual and textual representations (encoded by CLIP) as concept-to-image relevancy. "Naïve sampling" denotes selecting textual concepts based solely on relevancy before pre-training.

| Model | Expansion | | Ranking | | Sampling | | Average mIoU(%) |
|---|---|---|---|---|---|---|---|
| | *lang-driven* | *vision-driven* | *naïve* | *text-to-vision-guided* | *naïve* | *cluster-guided* | |
| Baseline [43] | | | | | | | 8.2 |
| #1 | ✓ | | ✓ | | ✓ | | 9.9 (1.7 ↑) |
| #2 | | ✓ | ✓ | | ✓ | | 10.3 (2.1 ↑) |
| #3 | | ✓ | | ✓ | ✓ | | 12.4 (4.2 ↑) |
| #4 (*Full* CoCu) | | ✓ | | ✓ | | ✓ | 13.1 (4.9 ↑) |

## 4.4 Analysis

**Ablation Study.** We further assess the effectiveness of each module in CoCu, which includes *vision-driven expansion*, *text-to-image-guided ranking* and *cluster-guided sampling*. Specifically, we pre-train five models with the combination of these modules or their alternative strategies, namely: 1) Baseline model of GroupViT, which is pre-trained without involving concept curation. 2) Model #1, which utilizes language-driven expansion, naïve ranking, and naïve sampling (Caption Curation in Sec. 3.2). 3) Model #2, which replaces language-driven expansion with vision-driven expansion on top of Model #1. 4) Model #3, which incorporates text-to-image-guided ranking on top of Model #2. And 4) the full CoCu Model #4, which combines vision-driven expansion, text-to-image-guided ranking, and cluster-guided sampling in pre-training. We report the average segmentation performance of these models across the eight datasets used previously (as shown in Table 1 and Table 2). Detailed illustrations of implementations are provided in appendix.

As Tab. 3 shows, the simplest strategy of *language-driven* in model #1 improves average mIoU by 1.7%, which comes from stronger vision-language correlation in pre-training data enhanced by direct text retrieval. Next, replacing direct text retrieval with *vision-driven expansion* in model #2 brings an additional performance boost, highlighting its significance in capturing unbiased semantics. Furthermore, incorporating *text-to-vision-guided ranking* in Model #3 brings another noticeable performance gain, underscoring the importance of measuring concept-to-image relevancy. Finally, we upgrade the sampling strategy from the naïve one that solely relies on relevancy to *cluster-guided sampling*, and build model #4 with the full CoCu, which provides more diverse semantic information in each pre-training step, ultimately leading to the best zero-shot transfer performance for semantic segmentation.

**Zero-Shot Classification**. In addition to its application in zero-shot segmentation, CoCu can also be used to improve zero-shot classification. Following the previous study [43], we evaluate CoCu and compare it with GroupViT on the ImageNet-1K dataset [13]. As shown in Table 4, CoCu exhibits significant performance gains over GroupViT, demonstrating its superiority in bridging semantic gaps across tasks and achieving improved zero-shot classification results.

Table 4: **Zero-shot classification on ImageNet-1K**. Acc@1 and Acc@5 denote top-1 and top-5 accuracy, respectively.

| Method | Pre-training data | Zero-shot | |
|---|---|---|---|
| | | Acc@1(%) | Acc@5(%) |
| GroupViT | C12 | 34.9 | 63.3 |
| CoCu (ours) | C12 | **38.4** (4.5 ↑) | **68.6** (5.3 ↑) |
| GroupViT | C3,C12,Y14 | 36.8 | 66.8 |
| CoCu (ours) | C3,C12,Y14 | **43.0** (6.2 ↑) | **73.7** (6.9 ↑) |

# 5  Conclusion

In this paper, we identify the issue of *semantic gap* in language-supervised semantic segmentation and explore how to bridge *semantic gaps* effectively. To achieve this, we design Concept Curation, a novel pipeline that resolves the issue by three consecutive stages: *vision-driven expansion*, *text-to-vision-guided ranking* and *cluster-guided sampling*. Extensive experiments demonstrate the superiority of our method for boosting language-supervised semantic segmentation across a bundle of pre-training sets and evaluation benchmarks. Looking ahead, we hope to extend the idea of concept curation to other computer vision tasks, including object detection and instance segmentation.

## Acknowledgments and Disclosure of Funding

This project is funded by the Ministry of Education Singapore, under the Tier-2 project scheme with a project number MOE-T2EP20220-0003.

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
