# Bridging Semantic Gaps for Language-Supervised Semantic Segmentation (Appendix)

## A   More Implementation Details

### A.1   Evaluation Benchmarks

As stated in section 4.1 of the manuscript, we follow [11] and predict only the foreground classes by thresholding the similarity between the embedding of output segments and text labels for 5 benchmarks, including Pascal VOC [4], Pascal Context [8], COCO [7], ImageNet-S-50, ImageNet-S-300 [5]. For other 3 evaluation benchmarks [3, 12, 1], we predict both foreground and background classes without thresholding. The number of evaluated classes and size of eight evaluation sets are listed in Tab. 1,

Table 1:  Details of evalution benchmarks.

| Dataset | Classes | Test Size |
|---|---|---|
| Pascal VOC [4] | 20 | 1,449 |
| ImageNet-S-50 [5] | 50 | 752 |
| Pacal Context [8] | 59 | 5,104 |
| COCO [7] | 80 | 5,000 |
| ImageNet-S-300 [5] | 300 | 4,097 |
| Cityscapes [3] | 19 | 500 |
| ADE20K [12] | 150 | 2,000 |
| COCO Stuff [1] | 171 | 5,000 |

### A.2   Comparing Methods

The comparing methods in Table 1 of the manuscript consist of two parts. The first part includes a fully supervised method (DeiT [10]) that learns from pixel-level annotations and finetunes the segmentation head on the training set of Pascal VOC [4] and Pascal Context [8]. The second part includes self-supervised methods (MoCo [6], DINO [2]) that pre-train visual representations by self-supervised learning [6, 2] and fine-tuning a segmentation head on pre-trained representations. Note the performance of all comparing methods are directly copied from a prior study [11] for fair comparison.

### A.3   Ablation Study

As claimed in the paper, we give the gist of modules in Table 3 of the manuscript and highlight their differences in Fig. 1 below. (a) Expansion: given an image as query, *language-driven expansion* directly retrieves potentially matched captions for later pre-training (please refer to section 3.2 of the manuscript for details) whereas *vision-driven expansion* retrieves image-text pairs and builds a concept archive; (b) Ranking: for computing relevancy of one concept $c_m$ to the query image, naïve

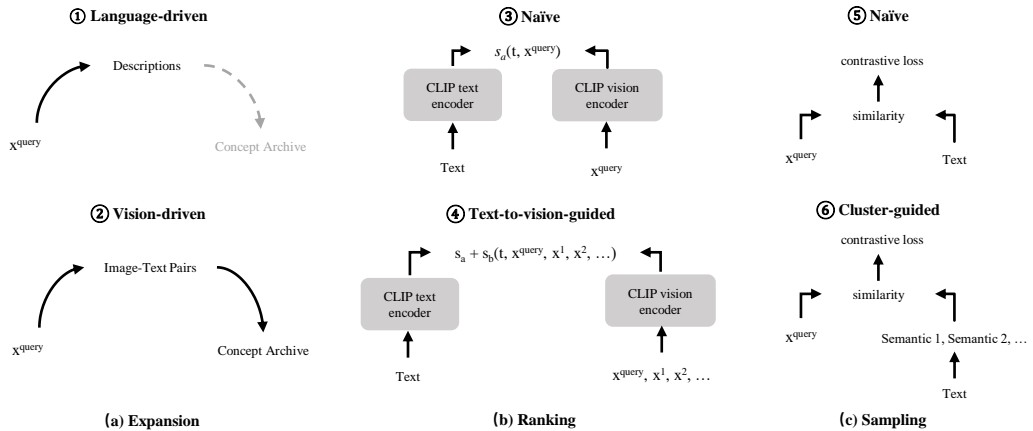

Figure 1: Illustration of modules in ablation study of the manuscript.

solution relies on $s_{c_m}^a$ solely but text-to-vision-guided ranking uses both $s_{c_m}^a$ and $s_{c_m}^b$ ($s_{c_m}^a$ and $s_{c_m}^b$ are computed as in equations (6) and (7) in the manuscript); (c) Sampling: cluster-guided strategy partitions the archive to semantic clusters before pre-training. Tab. 2 of this appendix lists the 4 ablation models in Table 3 of the manuscript and the corresponding combinations of the 6 modules in Fig. 1.

Table 2: Correspondence detail.

| Model | Module Combination |
|---|---|
| Baseline | N.A. |
| #1 | ①,③,⑤ |
| #2 | ②,③,⑤ |
| #3 | ②,④,⑤ |
| #4 | ②,④,⑥ |

# B  More Experiments

## B.1  Qualitative Visualization

We provide more activation maps of GroupViT [11] and CoCu by testing different textual concepts (of shown images) that are not captured in the corresponding captions. As shown in Fig. 2, the activation maps by GroupViT do not respond well at relevant image regions while CoCu activates at the right image regions and discriminates the textual concepts from other visual concepts in the images effectively. The performance difference is largely attributed to the concept curation in CoCu which captures the missing visual concepts and encodes them into pre-trained representations successfully.

As stated in Section 3.3 of the manuscript, the pre-trained CoCu models are more robust to changes of expressions of the same semantics (e.g., from "dog" to "kuvasz", "car" to "race car", etc.). As Fig. 3 shows, GroupViT behaves differently under the presence of expression changes while CoCu produces more consistent activation. The robustness to expression changes is largely attributed to two factors: 1) CoCu captures rich textual concepts that contain different expressions of the same semantics; 2) CoCu feeds semantics (as compared to textual concepts) into pre-training by selecting different expressions of the same semantics randomly.

## B.2  Parameter Learning

We study how parameter $N$ (i.e., the number of retrieved image-text pairs in expansion) affects pre-training and zero-shot transfer of pre-trained models. As described in Section 4.1, we carry

out pre-training on CC3M [9], evaluate over validation sets of eight benchmarks, and report the average mIoUs. We set $N$ at 8, 16, and 32 during expansion and report the performance of pre-trained segmentation models in Tab. 3. We can see that curation with 8 retrieved image-text pairs achieves slightly downgraded performance. While $N$ increases, the performance improves gradually and the best pre-training is obtained with 32 image-text pairs with an average mIoU of 13.1%.

Table 3: **Parameter learning**. $N$ denotes number of image-text pairs retrieved during expansion.

| Method | Retrieved Image-Text Pairs | Average mIoU (%) |
|---|---|---|
| GroupViT [11] | - | 8.2 |
| CoCu | 8 | 10.5 |
| | 16 | 12.9 |
| | 32 | 13.1 |

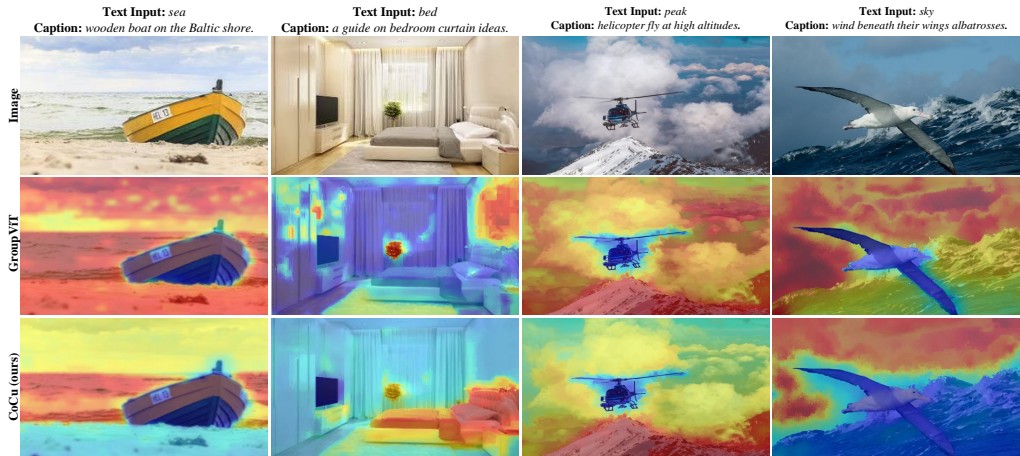

Figure 2: **GroupViT against CoCu on activation heatmaps**. Best viewed in color.

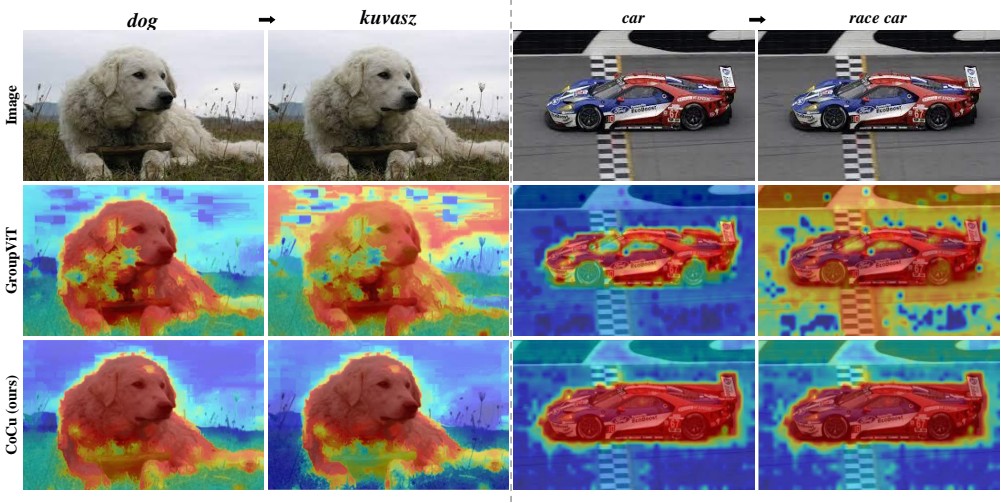

Figure 3: **CoCu is robust to changes of expressions of the same semantics**. Best viewed in color.

## C  Broader Impact

Pre-training large models on massive data may have broader societal impacts. Despite zero-shot segmentation performance on vast range of evaluation benchmarks, the pre-trained segmentors may encode undiscovered biases and stereotypes. Such models learnt on large-scale datasets should be checked before used for specific purposes, for instance, video surveillance or autonomous driving.