# OpenReview forum: "Rewrite Caption Semantics: Bridging Semantic Gaps for Language-Supervised Semantic Segmentation"
_NeurIPS.cc/2023/Conference — NeurIPS 2023 poster_

### Official Review · Reviewer_wvpf · 2023-07-04

**Soundness:** 3 good
**Presentation:** 3 good
**Contribution:** 3 good
**Rating:** 6
**Confidence:** 3

**Summary:**

This paper first points out the semantic gap problem of the existing language-supervised semantic segmentation method. It shows that not all visual elements are included in the corresponding language annotations. Then the paper proposes Concept Curation (CoCu), which includes Vision-driven Expansion, Text-to-Vision-Guided Ranking, and Cluster-guided Sampling strategies to solve the semantic gap problem. The experiments show the effectiveness of the CoCu.

**Strengths:**

1. The semantic gap problem sounds very reasonable and significant for language-supervised semantic segmentation.
2. The proposed method CoCu could alleviate the semantic gap problem to some extent.
3. The experiments and ablation study are comprehensive.

**Weaknesses:**

1. The authors should summarize the Sec. 2.3 at the beginning or the end of this section. For example, in general, Sec. 2.3 provides a method to find better class candidates for the loss in Sec. 2.1. Otherwise it confuses the readers about the ultimate goal of Sec. 2.3.
2. The overall pipeline of CoCu is a kind of dataset pre-processing method, and it is complicated. Is CoCu done online or offline? If online, how much time does it cost to find the final class candidates?

**Questions:**

None

---

> ### Author Rebuttal · Authors · 2023-08-10
>
> We thank for your valuable suggestions and response as follows for your concern.
>
> **Q1: Writing in Sec. 3.3?**
>
> Thanks for pointing it out. As suggested, we will clarify the motivation of our method at the beginning of Section 3.3 as well as its relation with Section 3.1.
>
> **Q2: Online or Offline? Time Cost?**
>
> We implement CoCu in an offline manner so there is no additional online time cost as compared to baseline method. For offline computation, performing Concept Curation on CC3M [36] dataset roughly takes 1.6 hours (which is nearly 8% of pre-training time following our setting) but brings +4.9% performance gains. Please refer to a table below for more details on time cost.
>
> |**Method**|**Step**|**Computation**|**Operation**|**Time Cost (hrs)**|**Average mIoU (%)**|
> |:-----|:-----|:-----|:-----|:-----|:-----|
> | GroupViT [43] | 1 (final step) | online | pre-train | 20.0 | 8.2 |
> | CoCu | 1 | offline | inference | 0.3 |  |
> | CoCu | 2 | offline | build index | 0.1 |  |
> | CoCu | 3 | offline | curation | 1.2 |  |
> | CoCu | 4 (final step) | online | pre-train | 20.0 | 13.1 (+4.9) |

---

> > ### Comment · Reviewer_wvpf · 2023-08-14
> > **Response to the rebuttal**
> >
> > Thanks for the answer. I keep my rating.

---

### Official Review · Reviewer_iruH · 2023-07-04

**Soundness:** 4 excellent
**Presentation:** 2 fair
**Contribution:** 3 good
**Rating:** 6
**Confidence:** 4

**Summary:**

This paper proposed a novel data curation/argumentation process, named Concept Curation (CoCu), for language-supervised semantic segmentation. In the setting of language-supervised semantic segmentation (e.g. GroupViT), the network is trained with image-text contrastive loss on large-scale image-text pairs. Authors identified several issues in the vanilla data distribution of the original contrastive learning, e.g. semantic gap, and semantic bias. The proposed CoCu mitigates these issues and improves over prior work GroupViT under a controlled setting, showing a faster convergence rate and high accuracy.

**Strengths:**

1. In the quantitative evaluation, authors re-implemented GroupViT and compare CoCu and GroupViT under the controlled experiment setting, i.e. 1024 batch size. The authors also report the accuracies on additional evaluation datasets, IN50, IN300, Cityscapes, etc. CoCu outperforms GroupViT on all the datasets by a margin. Although CoCu doesn't achieve state-of-the-art results on some datasets, it doesn't degrade the effectiveness of the method.
2. The proposed method is well-motivated. The web-crawled image-text pairs are indeed quite noisy. And GroupViT is also known to be bad at segmenting background classes like grass and sky. Moreover, from the visualization in Figure 3(b), the CoCu shows that could focus on background grasses instead of the foreground fox.
3. In the Table 2 ablation study, when only trained on CC3M, CoCu improves over GroupViT by a large margin on Pascal VOC. It is a very interesting result and justifies that CoCu speeds up the convergence.

**Weaknesses:**

1. Insufficient qualitative comparison. Since one major claim of CoCu is that, compared with vanilla contrastive loss used in GroupViT, the proposed dataset curation mitigates semantic cap and semantic bias issues. So besides quantitative evaluation metric mIoU, more visualizations compared with GroupViT are expected. I would suggest authors add more visualizations in the supplementary materials if there is no space left in the main submission.
2. In the ablation table 3, the authors studied the effects of different components of CoCu. But the average mIoU may not be an insightful metric. Since different datasets have very different category vocabulary, authors may try to include a detailed ablation study table in the supplementary material and elaborate more on the results.

**Questions:**

1. The boundary between the semantic gap and semantic bias is not that clear. At least from Figure 1, it's hard to tell the major differences between them. Maybe authors could define them more rigorously and provide more examples about them.
2. How long does it take to process the dataset? It would be informative for other readers who want to reproduce or follow up on the curation process.
3. In Table 2, CoCu also showed significant improvement on ImageNet50 and ImageNet300. But for these two datasets, the vocabularies are mainly foreground, it would be very interesting to show why CoCu could improve on these datasets as well.

**Limitations:**

Yes.

---

> ### Author Rebuttal · Authors · 2023-08-10
>
> We thank you for your value of our efforts and helpful suggestions. Please check our clarification below regarding your concerns.
>
> **Q1: More qualitative comparison.**
>
> Please refer to the Appendix Figure 2 for additional qualitative comparisons (heatmap). The figure shows that CoCu learns visual concepts (especially not captured by captions) better than the baseline GroupViT (as presented in manuscript Figure 4). As suggested, we will provide more examples presenting better convergence to missing concepts in updated Appendix.
>
> **Q2: Ablation study table.**
>
> Please find a more detailed ablation study table below for different evaluation datasets, where "baseline", "#1", "#2", "#3" and "#4" refer to the same experiments as in manuscript Table 3. As suggested, we will include this table into the Appendix for reference.
>
> |**Model** | **PVOC** | **PCON** | **COCO** | **IN-S-50** | **IN-S-300** | **CITY** | **ADE** | **STUF** | **Average** |
> |:-----|:-----|:-----|:-----|:-----|:-----|:-----|:-----|:-----|:-----|
> | Baseline | 15.5 | 10.4 | 6.5 | 10.2 | 2.9 | 8.1 | 4.4 | 7.7 | 8.2 |
> | #1 | 20.7 | 10.4 | 8.2 | 13.9 | 4.6 | 8.3 | 4.9 | 8.0 | 9.9 |
> | #2 | 22.8 | 11.4 | 8.6 | 13.5 | 5.1 | 8.3 | 5.8 | 7.0 | 10.3 |
> | #3 | 29.2 | 13.0 | 10.2 | 18.8 | 6.3 | 7.5 | 5.9 | 8.3 | 12.4 |
> | #4 | 30.6 | 13.9 | 10.8 | 19.3 | 7.3 | 8.2 | 6.1 | 8.5 | 13.1 |
>
> **Q3: Difference between semantic gap and semantic bias?**
>
> We further clarify these two concepts as suggested.
>
> We refer to semantic gap as a problem underlying in **pre-training data**. In web-crawled image-text pairs, it is prevalent that texts (i.e., captions) do not capture comprehensive visual concepts in their paired images, yielding semantic gap between the paired texts and images.
>
> In comparison, we refer to **semantic bias** as a problem in **pre-trained vision-language models**. We notice that pre-trained vision-language models often retrieves salient visual concepts in images when taking images as inputs and attempting to retrieve texts, hence demonstrating certain bias toward salient visual concepts.
>
> We conjecture that semantic bias in large-scale vision-language models originates from the semantic gap in pre-training data they used to pre-train the large-scale models. For semantic bias in CLIP models, semantic gap lies in the noisy WIT400M image-text pairs.
>
> We will further define and clarify these two concepts in the updated manuscript.
>
> **Q4: Time cost to process the dataset?**
>
> For reference, processing CC3M [36] dataset with our method takes roughly 1.6 hours (8% of pre-training time) including fast inferencing images and texts, building search index, and performing concept curation on all image-text pairs. Please refer to the table below for more details on time cost.
>
> |**Method**|**Step**|**Computation**|**Operation**|**Time Cost (hrs)**|**Average mIoU (%)**|
> |:-----|:-----|:-----|:-----|:-----|:-----|
> | GroupViT [43] | 1 (final step) | online | pre-train | 20.0 | 8.2 |
> | CoCu | 1 | offline | inference | 0.3 |  |
> | CoCu | 2 | offline | build index | 0.1 |  |
> | CoCu | 3 | offline | curation | 1.2 |  |
> | CoCu | 4 (final step) | online | pre-train | 20.0 | 13.1 (+4.9) |
>
> **Q5: Why CoCu is also effective for foreground regions?**
>
> In Figure 3 of the submitted Appendix, we provided qualitative comparisons between the baseline method GroupViT [43] and our CoCu for segmenting ImageNet-S foreground categories. We showcased that **with CoCu pre-training, the segmentator is more robust to  changes of expressions of the same semantics in captions**, e.g., from "dog" to "kuvasz", demonstrating that CoCu leads to more effective and generalized representation learning over foreground categories.
>
> We include a new example for segmenting foreground regions in Figure 1 in the attached **"global-response.pdf''**. Specifically, in this figure, we segment an image captioned *helicopter fly at high altitudes* with both GroupViT [43] (first row) and CoCu (second row) pre-trained models. When we feed different text inputs (i.e., *cloud*, *helicopter* and *mountain*) to both segmentors and visualize their activations, it is clear to see that GroupViT [43] activated on *helicopter* region (highlighted with red box) with all given text inputs (i.e., *helicopter*, *cloud*, *mountain*). In such a case, the pre-trained segmentor struggled to discriminate *helicopter* from other concepts (*cloud*, *mountain*). As a comparison, CoCu successfully captured corresponding regions when given different text inputs. This is because the  pre-training in the baseline GroupVIT captures insufficient textual concepts in its visual representations, thus is easily confused by text inputs. In contrast, CoCu segmentor learns significantly more comprehensive concepts in pre-training by bridging semantic gaps and is more robust to variations of semantic context, thus localizing each text input more accurately. This further showcased that **our CoCu facilitates discriminating different contexts in the same images**, leading to better foreground segmentation on ImageNet-S-50 and ImageNet-S-300. We thank you for your valuable comments and will include this analysis in the updated appendix.

---

### Official Review · Reviewer_griQ · 2023-07-05

**Soundness:** 2 fair
**Presentation:** 3 good
**Contribution:** 2 fair
**Rating:** 6
**Confidence:** 4

**Summary:**

This paper targets learning unsupervised semantic segmentation from image-text pairs. The authors specifically address the issue of training data quality in previous method Group-ViT and propose an approach to enhance the captions with additional visual concepts through an automated pipeline. The paper demonstrates the effectiveness of this data filtering pipeline by evaluating it on official segmentation benchmarks.

**Strengths:**

- The semantic gap problem the authors studied is intresting and meaning full.
- The writing is good and the paper is easy to understand.

**Weaknesses:**

The proposed data filtering pipeline relies on the CLIP model for collecting visually similar samples. However, the CLIP model has been trained on a much larger scale, with hundreds of millions of image-text pairs, whereas the training data used in this paper is relatively smaller. Does it work by distilling the CLIP model? This raises concerns about **the effectiveness of the pipeline when scaling up to larger training data sizes**. It would be helpful to explore **alternative self-supervised methods such as DINO, MAE, or Group-ViT itself to replace the CLIP model** in the pipeline and evaluate its performance.

**Questions:**

See weakness.

**Limitations:**

NaN

---

> ### Author Rebuttal · Authors · 2023-08-10
>
> We thank you for your valuable suggestion. Below please find our clarification regarding your concern.
>
> **Q1: Does it work by distilling CLIP Model? Replace CLIP with GroupViT?**
>
> No, CoCu does not rely on vast knowledge encoded in CLIP [32] but more on our novel pipeline design. To verify this, we replace CLIP with GroupViT as our curation model to perform CoCu. The quantitative comparison with our baseline is presented in the Table below. As presented, performing CoCu with GroupViT brings decent performance gains (+3.9%) over the baseline method, which is comparable with that of performing CoCu by CLIP (+4.9%). Hence, the performance gain is largely attributed to our CoCu design, though employing the powerful CLIP further boosts the performance
>
> | **Curation Model** | **Backbone** | **PVOC** | **PCON** | **COCO** | **IN-S-50** | **IN-S-300** | **CITY** | **ADE** | **STUF** | **Average** |
> |:-----|:-----|:-----|:-----|:-----|:-----|:-----|:-----|:-----|:-----|:-----|
> | - | - | 15.5 | 10.4 | 6.5 | 10.2 | 2.9 | 8.1 | 4.4 | 7.7 | 8.2 |
> | GroupViT [43] (newly added) | ViT-S | 26.1 | 12.3 | 10.4 | 17.8 | 6.9 | 7.6 | 6.8 | 8.7 | 12.1 (+3.9) |
> | CLIP [32] | ViT-B/16 | 30.6 | 13.9 | 10.8 | 19.3 | 7.3 | 8.2 | 6.1 | 8.5 | 13.1 (+4.9) |
>
> **Q2: Scaling up to larger training size?**
>
> We did not perform large-scale experiments due to resource limitations. For the pre-training scale in our manuscript Table 1, it takes around 9 days with Tesla V100 GPUs and the global batch size of 1024. Increasing pre-training scales further will lead to much longer training time. Nevertheless, Table 2 of the manuscript shows how CoCu behaves under various pre-training scales, indicating that CoCu's superior performance is quite tolerant to the pre-training scale.
>
> **Q3: Replace CLIP with Self-Supervised Models?**
>
> We should clarify that CoCu is not applicable to DINO or MAE as CoCu takes texts as inputs, which is not supported by self-supervised models that handle image inputs solely. We surely agree that it is very meaningful to explore what role self-supervised models could play in bridging semantic gaps, and we will study it in the future work.

---

### Official Review · Reviewer_e4vq · 2023-07-06

**Soundness:** 2 fair
**Presentation:** 3 good
**Contribution:** 2 fair
**Rating:** 5
**Confidence:** 5

**Summary:**

Current VLMs suffer from a noticeable semantic gap between visual and textual modalities since many visual concepts present in images are easily missed in their paired captions. This work proposes Concept Curation (CoCu), a pipeline that leverages CLIP to compensate for the missing semantics. For each image-text pair, CoCu establishes a concept archive that maintains potential visually-matched concepts using vision-driven expansion and text-to-vision-guided ranking. This approach enables the identification of relevant concepts through cluster-guided sampling, which are then fed into the pre-training process. As a result, CoCu bridges the gap between visual and textual semantics. Extensive experiments conducted on eight segmentation benchmarks demonstrate that CoCu achieves exceptional zero-shot transfer performance and significantly enhances the language-supervised segmentation baseline by a substantial margin. These results underscore the importance of closing the semantic gap in pre-training data. The code for CoCu will be made available to the research community.

**Strengths:**

1. Overall, the paper is well-written, and the study of this work has a good motivation.
2. The mechanism of CoCu is intuitive and easy to follow.
3. CoCu demonstrates significant empirical results: it outperforms GroupViT (re-implemented baseline) by 4.6% mIoU in average on eight popular semantic segmentation benchmarks.
4. CoCu also yield good qualitative results, with more accurate and smooth masks than its baseline.

**Weaknesses:**

1. Despite the fact that CoCu indeed enriches visual concepts during VL pre-training, the cost seems to be heavy. During training, a CLIP model is employed to perform the key components of CoCu such as vision-driven expansion and text-to-vision guided ranking, which introduces a lot of additional computation. Thus, the comparison to your baselines such as GroupViT might not be strictly fair. A detailed comparison of training time or FLOPS should be presented to support the effectiveness of CoCu.

2. My bigger concern lies in how CoCu relies on the pre-trained CLIP, i.e., if using a weaker CLIP model for CoCu, how will the performance change? If CoCu is not sensitive to the CLIP's performance, you can directly use GroupViT to perform vision-driven expansion and text-to-image-guided ranking, so that no additional parameters will be introduced. As shown in Table 4, GroupViT obtains acceptable zero-shot classification results. However, if CoCu highly relies on a strong CLIP, your contributions might also be challenged.

**Questions:**

1. In table 1, are both CoCu and GroupViT trained for 30 epochs?
2. With the concern of Weakness 1, I am also wondering what if you change the x-axis from #epochs to actual training time in Figure 3? Will CoCu still converge faster than GroupViT?

**Limitations:**

As CoCu basically follows the architecture of GroupViT, it has the same limitation that for each image, the number of semantic concepts segmented by CoCu/GroupViT has a maximum of the number of group tokens fed to the vision encoder. Thus, given a high-resolution image with many semantic regions, these methods might fail to generate accurate segmentation masks.

---

> ### Author Rebuttal · Authors · 2023-08-10
>
> Thank you for your comments. We appreciate the value you see in our innovative motivation and competitive experimental results. Below, please find clarifications regarding your concerns below.
>
> **Q1: In Table 1, are both CoCu and GroupViT trained for 30 epochs?**
>
> Yes, in manuscript Table 1, pre-training configurations of GroupViT [43] and CoCu remain the same to ensure a fair comparison.
>
> **Q2: Change the x-axis from #epochs to actual training time in Figure 3?**
>
> We would clarify that CoCu is implemented in an offline manner, and its does not introduce much overhead in processing. Take the experiment on CC3M [36] dataset as an example. CoCu takes 1.6 hours in total as detailed in the table below, and the rest pre-training is similar to the pre-training in GroupViT [43] which takes around 20 hours. Hence, CoCu introduces around 8% computational overhead. We provide the time cost of each operation in Table below.
>
> |**Method**|**Step**|**Computation**|**Operation**|**Time Cost (hrs)**|**Average mIoU (%)**|
> |:-----|:-----|:-----|:-----|:-----|:-----|
> | GroupViT [43] | 1 (final step) | online | pre-train | 20.0 | 8.2 |
> | CoCu | 1 | offline | inference | 0.3 |  |
> | CoCu | 2 | offline | build index | 0.1 |  |
> | CoCu | 3 | offline | curation | 1.2 |  |
> | CoCu | 4 (final step) | online | pre-train | 20.0 | 13.1 (+4.9) |
>
> Despite the 8% additional offline time cost, CoCu brings remarkable performance gains in comparison to the baseline GroupViT [43] (+4.9% over 8 benchmarks). This supports the significance of bridging semantic gaps in language-supervised segmentation, and offers flexibility while tackling different tasks with different requirements.
>
> As suggested, we will update a new loss curve that uses actual time as x-axis in manuscript Figure 3. The time usage will include both offline operations and online pre-training.
>
> **Q3: For curation model, replace CLIP with a weaker CLIP model and GroupViT?**
>
> Technically, CoCu could leverage any vision-language model to perform curation, bridge semantic gaps and facilitate language-supervised segmentation. However, we would highlight that a strong curation model (e.g., CLIP [32]) is not a necessity to achieve decent performance gains in downstream datasets.
>
> To verify this, we follow your suggestion and replace CLIP model (CLIP ViT-B/16) we used in manuscript with a weaker CLIP model (CLIP ViT-B/32) to perform CoCu, followed by pre-training a segmentor on CC3M [36]. The experiment shows that CoCu with CLIP ViT-B/32 brings comparable performance gains over baseline (8.2%) as compared to that of using CLIP ViT-B/16 (12.8% v.s. 13.1%). This demonstrates that CoCu can work with different CLIP models with consistent performance gains.
>
> To ensure fairness to baseline method [43], we further replace CLIP ViT-B/16 with GroupViT [43] as our curation model to perform CoCu. As presented in Table below, performing CoCu with GroupViT still brings clear improvement (+3.9%) over baseline method. This indicates the significant performance gain is largely attributed to our novel pipeline design, though employing the powerful CLIP further boosts the performance. Please note that the retrieved results vary depending on different language-vision models, leading to fluctuating pre-training and segmentation results. Nevertheless, the enhancements across all models remain noteworthy, underscoring our pipeline’s inherent capacity for generalization and improvement.
>
> | **Curation Model** | **Backbone** | **PVOC** | **PCON** | **COCO** | **IN-S-50** | **IN-S-300** | **CITY** | **ADE** | **STUF** | **Average** |
> |:-----|:-----|:-----|:-----|:-----|:-----|:-----|:-----|:-----|:-----|:-----|
> | - | - | 15.5 | 10.4 | 6.5 | 10.2 | 2.9 | 8.1 | 4.4 | 7.7 | 8.2 |
> | GroupViT [43] (newly added) | ViT-S | 26.1 | 12.3 | 10.4 | 17.8 | 6.9 | 7.6 | 6.8 | 8.7 | 12.1 (+3.9) |
> | CLIP [32] (newly added) | ViT-B/32 | 27.4 | 14.8 | 10.6 | 16.7 | 6.3 | 10.4 | 6.3 | 9.9 | 12.8 (+4.6) |
> | CLIP [32] | ViT-B/16 | 30.6 | 13.9 | 10.8 | 19.3 | 7.3 | 8.2 | 6.1 | 8.5 | 13.1 (+4.9) |

---

> > ### Comment · Reviewer_e4vq · 2023-08-20
> > **Response to authors' rebuttal**
> >
> > Thank you for your rebuttal and additional experiments. My primary concerns are addressed, and I would like to raise my rating to borderline acceptance.

---

### Author Rebuttal · Authors · 2023-08-10

Please check global_response.pdf for our new Figures.

---

### Comment · Area_Chair_Dbh4 · 2023-08-18

Hi Authors,

Thanks a lot for your rebuttal. We are urging all the reviewers to respond and will take your input into consideration as we make the final recommendation. Thanks!

Best,

---

### Decision · Program_Chairs · 2023-09-21

**Decision:**

Accept (poster)

**Comment:**

The paper received 6/6/5/6 ratings. Reviewers acknowledge the problem of filling semantic gaps for language supervised segmentation is interesting, well motivated, straightforward and the paper is well presented. While some concerns such as how the CLIP model affects the performance of the method were raised, those were successfully addressed through rebuttal. The AC recommends to accept the paper.